# GPUDrive: Data-driven, multi-agent driving simulation at 1 million FPS

**Saman Kazemkhani**[*†1]         **Aarav Pandya**[*†1]         **Daphne Cornelisse**[*†1]

**Brennan Shacklett**[2]                    **Eugene Vinitsky**[1]

## Abstract

Multi-agent learning algorithms have been successful at generating superhuman planning in various games but have had limited impact on the design of deployed multi-agent planners. A key bottleneck in applying these techniques to multi-agent planning is that they require billions of steps of experience. To enable the study of multi-agent planning at scale, we present GPUDrive. GPUDrive is a GPU-accelerated, multi-agent simulator built on top of the Madrona Game Engine capable of generating over a million simulation steps per second. Observation, reward, and dynamics functions are written directly in C++, allowing users to define complex, heterogeneous agent behaviors that are lowered to high-performance CUDA. Despite these low-level optimizations, GPUDrive is fully accessible through Python, offering a seamless and efficient workflow for multi-agent, closed-loop simulation. Using GPUDrive, we train reinforcement learning agents on the Waymo Open Motion Dataset, achieving efficient goal-reaching in minutes and scaling to thousands of scenarios in hours. We open-source the code and pre-trained agents at

www.github.com/Emerge-Lab/gpudrive

## 1 Introduction

Multi-agent learning has been impactful across a wide range of fully cooperative and zero-sum games (Cui et al., 2023; Wurman et al., 2022; Pérolat et al., 2022; Silver et al., 2017; Jaderberg et al., 2018; Bakhtin et al., 2023). However, its impact on multi-agent planning for settings that mix humans and robots has been muted. In contrast to the ubiquity of multi-agent learning-based agents in zero-sum games, multi-agent planners for most practical robotic systems are not derived from the output of game-theoretically sound learning algorithms. While it is hard to characterize the space of deployed planners since many of them are proprietary, the majority likely use a mixture of collected data for the prediction of human motion and hand-tuned costs. These are then fed into a cost-based trajectory optimizer or into a planner based on imitation learning (Bronstein et al., 2022; Lu et al., 2023). This approach has been highly effective in scaling up real-world autonomy but can struggle with reasoning about long-term behavior, contingency planning, and interaction with humans in rare, complex scenarios.

The divergence in preferred technique between these two domains is partially the outcome of two distinct, challenging components of real-world multi-agent planning. First, unlike zero-sum games, it is necessary to play a human-compatible strategy that is difficult to identify without data. Second, generating the billions of samples needed for multi-agent learning algorithms is difficult with existing simulators. The former challenge is difficult for multi-agent learning since there is not a clear equilibrium concept that algorithms should be pursuing. The latter problem is a challenge for simulators since it is difficult to simulate embodied multi-agent environments at appropriately high rates.

---

[1]NYU Tandon School of Engineering

[2]Stanford University

[*]These authors contributed equally to this work

[†]Corresponding authors: skazemkhani@gmail.com, pandya.aarav.97@gmail.com, cornelisse.daphne@nyu.edu.

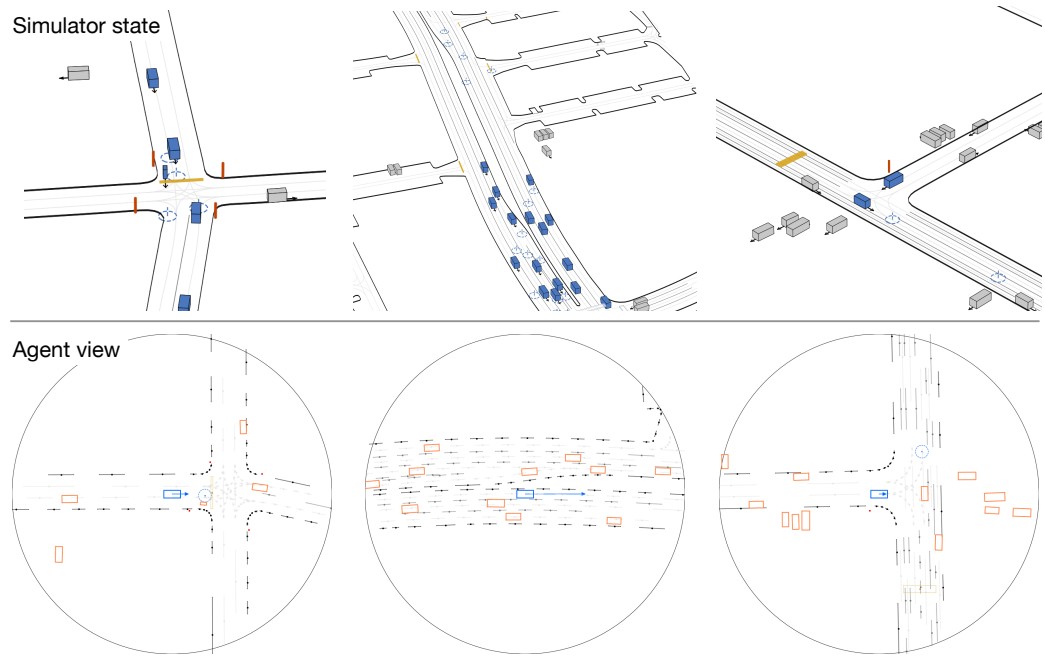

Figure 1: **Extremely fast multi-agent simulation with GPUDrive**. *Top*: Bird's-eye view of Waymo Open Motion Dataset scenarios in GPUDrive, with boxes marking controlled agents and circles denoting their goals. *Bottom*: Corresponding agent views, centered on one agent. Observations can be easily configured based on the user's objectives. Here, agents are provided with a scene view through a relative coordinate frame. Shown are nearby road points within a configurable radius (set to 50 meters) and the relative positions of other agents in the scene.

To address these challenges and unlock multi-agent learning as a tool for generating capable self-driving planners, we introduce GPUDrive. GPUDrive is a simulator intended to mix real-world driving data with simulation speeds that enable the application of sample-inefficient but effective RL algorithms to the design of autonomous planners. GPUDrive runs at over a million steps per second on both consumer-grade and datacenter-class GPUs and has a sufficiently light memory footprint to support hundreds to thousands of simultaneous worlds (environments) with hundreds of agents per world. GPUDrive supports the simulation of a variety of sensor modalities, from LIDAR to a human-like view cone, enabling GPUDrive to be used for studying the effects of different sensor types on resultant agent characteristics. Finally, GPUDrive takes in driving logs and maps from existing self-driving datasets, enabling the mixing of tools from imitation learning with reinforcement learning algorithms. This enables the study of both the development of autonomous vehicles and the learning of models of human driving, cycling, and walking behavior.

Our contributions are:

- We provide a multi-agent, GPU-accelerated, and data-driven simulator that runs at over a million steps per second (Section 4.1). Our simulator provides a testbed for:

  1. Investigating the capability of learning algorithms to solve challenges related to self-play or autonomous coordination.
  2. Researching the effects of limited or human-like perception on agent behavior.

- We provide `gymnasium` (Towers et al., 2024) environments in both `torch` and `jax` that can be easily configured with standard open-source multi-agent RL and imitation learning libraries. Additionally, we open-sourced two policy-gradient training loops that can be readily used to develop agents.

- We release implementations of tuned RL algorithms that can process 30 million steps of experience per hour on consumer-grade GPUs. These can be used to train 95% goal-reaching

agents across 1000 different multi-agent scenarios in 15 hours on relatively accessible hardware.

- We open-source these strong driving baseline agents that achieve 95% of their goals on a subset of scenes. These are integrated into the simulator so that the simulator comes with default, reactive agents.

## 2 RELATED WORK

**Frameworks for batched simulators.** There are various open-source frameworks available that support hardware-accelerated reinforcement learning environments. These environments are generally written directly in an acceleration framework such as Numpy (Harris et al., 2020), Jax (Bradbury et al., 2018), or Pytorch (Ansel et al., 2024; Makoviychuk et al., 2021; Panerati et al., 2021). In terms of multi-agent accelerated environments, standard benchmarks include JaxMARL(Rutherford et al., 2023), Jumanji (Bonnet et al., 2024), VMAS (Bettini et al., 2022) Gigastep (Lechner et al., 2024) which primarily feature fully cooperative or fully competitive tasks. Each benchmark requires the design of custom accelerated structures per environment. In contrast, GPUDrive focuses on a mixed motive setting and is built atop Madrona, an extensible ECS-based framework in C++, enabling GPU acceleration and parallelization across environments (Shacklett et al., 2023). Madrona comes with vectorization of key components of embodied simulation such as collision checking and sensors such as LIDAR. GPUDrive can support hundreds of controllable agents in more than 100,000 distinct scenarios, offering a distinct generalization challenge and scale relative to existing benchmarks. Moreover, GPUDrive includes a large dataset of human demonstrations, enabling imitation learning, inverse RL, and combined IL-RL approaches.

**Simulators for autonomous driving research and development.** Table 1 shows an overview of current simulators used in autonomous driving research. The purpose of GPUDrive is to facilitate the systematic study of behavioral, coordination, and control aspects of autonomous driving and multi-agent learning more broadly. As such, visual complexity is reduced, which differs from several existing simulators, which (partially) focus on perception challenges in driving (Dosovitskiy et al., 2017; Cai et al., 2020). Driving simulators close to GPUDrive in terms of either features or speed include MetaDrive (Li et al., 2022), nuPlan (Caesar et al., 2021), Nocturne (Vinitsky et al., 2022), and Waymax (Gulino et al., 2024) which all utilize real-world data. Unlike MetaDrive and nuPlan, our simulator is GPU-accelerated. Like GPUDrive, Waymax is a JAX-based GPU-accelerated simulator that achieves high throughput through JIT compilation and efficient use of accelerators. With respect to Waymax, our simulator supports a wider range of possible sensor modalities (Section 3.2) including LIDAR and human-like views, can scale to nearly thirty times more worlds (see Section 4.1), and comes with performant reinforcement learning baselines. However, it does not currently come with reactive IDM agents like Waymax though it does come with pre-trained simulated agents based on RL policies.

**Driving agents in simulators and algorithms.** Existing simulators often feature baseline agents for interaction, such as *low-dimensional car following models* that describe vehicle dynamics through a limited set of variables or parameters (Kreutz & Eggert, 2021; Kesting et al., 2007; Treiber et al., 2000). Rule-based agents exhibit predetermined behaviors, like car-following agents (Gulino et al., 2024; Caesar et al., 2021; Lopez et al., 2018; Casas et al., 2010) such as the IDM model, or parameterized behavior agents like CARLA's TrafficManager (Dosovitskiy et al., 2017). Some simulators offer *recorded human driving logs* for interaction through replaying the human driving logs (Lu et al., 2023; Vinitsky et al., 2022; Gulino et al., 2024; Caesar et al., 2021). Additionally, certain simulators provide *learning-based agents*, leveraging reinforcement learning techniques (Li et al., 2022). In GPUDrive, we provide both human driving logs and high-performing reinforcement learning agents.

Table 1: Comparison of GPUDrive to related driving simulators. Columns represent whether the simulator supports multi-agent simulation, GPU acceleration, simulation of sensors such as LIDAR or human views, is built atop data, comes with existing driver models, and whether the agents are provided explicit goal points or waypoints along the way to the goal.

| Simulator | Multi-agent | GPU-Accel | Sensor Sim | Expert Data | Sim-agents | Routes / Goals |
|---|---|---|---|---|---|---|
| TORCS (Wymann et al., 2000) | | | ✓ | | ✓ | - |
| GTA V (Martinez et al., 2017) | | | ✓ | | | - |
| CARLA (Dosovitskiy et al., 2017) | | | ✓ | | ✓ | Waypoints |
| Highway-env (Leurent et al., 2018) | | | | | | - |
| Sim4CV (Müller et al., 2018) | | | ✓ | | | Directions |
| SUMMIT (Cai et al., 2020) | ✓ ($\geq 400$) | | ✓ | | ✓ | - |
| MACAD (Palanisamy, 2020) | ✓ | | ✓ | | ✓ | Goal point |
| SMARTS (Zhou et al., 2021) | ✓ | | | | | Waypoints |
| MADRaS (Santara et al., 2021) | ✓ ($\geq 10$) | | ✓ | | | Goal point |
| DriverGym (Kothari et al., 2021) | | | | ✓ | ✓ | - |
| VISTA (Amini et al., 2022) | ✓ | | ✓ | ✓ | | - |
| nuPlan (Caesar et al., 2021) | | | ✓ | ✓ | ✓ | Waypoints |
| Nocturne (Vinitsky et al., 2022) | ✓ ($\geq 128$) | | | ✓ | ✓ | Goal point |
| MetaDrive (Li et al., 2022) | ✓ | | ✓ | ✓ | ✓ | - |
| InterSim (Sun et al., 2022) | ✓ | | | ✓ | ✓ | Goal point |
| TorchDriveSim (Ścibior et al., 2021) | ✓ | ✓ | | | ✓ | - |
| BITS (Xu et al., 2023) | ✓ | | | ✓ | ✓ | Goal point |
| Waymax (Gulino et al., 2024) | ✓ ($\geq 128$) | ✓ | | ✓ | ✓ | Waypoints |
| **GPUDrive (ours)** | ✓ ($\geq 128$) | ✓ | ✓ | ✓ | ✓ | Goal point |

# 3 SIMULATION DESIGN

## 3.1 SIMULATION ENGINE

Learning to safely navigate complex scenarios in a multi-agent setting requires generating many billions of environment samples. To feed sample-hungry learning algorithms, GPUDrive is built on top of Madrona (Shacklett et al., 2023), an Entity-Component-State system designed for high-throughput reinforcement learning environments. In the Madrona framework, multiple independent worlds (each containing an independent number of agents[†]) are executed in parallel on an accelerator via a shared engine.

However, the simulation of driving poses a unique set of challenges which require careful technical design choices to solve. First, road objects, such as road edges and lane lines are frequently represented as polylines (i.e. connected sets of points). These polylines can consist of hundreds of points as they are sampled at every 0.1 meters, leading to even small maps having upwards of tens of thousands of points. This can blow up the memory requirements of each world as well as lead to significant redundancy in agent observations. Second, the large numbers of agents and road objects can make collision checking a throughput bottleneck. Finally, there is immense variability in the number of agents and road objects in a particular scene. Each world allocates memory to data structures that track its state and accelerate simulation code. Though independent, each world incurs a memory footprint proportional to the *maximum* number of agents across all worlds. In this way, the performance of GPUDrive is sensitive to the variation in agent counts across all the worlds in a batch.

These challenges are partially resolved via the following mechanisms. First, a primary acceleration data structure leveraged by GPUDrive is a Bounding Volume Hierarchy (BVH). The BVH keeps track of all physics entities and is used to easily exclude candidate pairs for collisions. This allows us to then run a reduced-size collision check on potential collision candidate pairs. The use of a BVH avoids invoking a collision check that would otherwise always be quadratic in the number of agents in a world. Secondly, we observed that a lot of the lines in the geometry of the roads are straight. This allows us to omit many intermediate points while only suffering a minor hit in the quality of the curves. We apply a polyline decimation algorithm (Visvalingham-Whyatt Algorithm) (Visvalingam & Whyatt, 1993) to approximate straight lines and filter out low-importance points in the polylines. With this modification, we can reduce the number of points by 10-15 times and significantly improve the step times while decreasing memory usage. Finally, rather than allocate memory for the maxi-

---

[†]In the Waymo Open Motion Dataset, an agent constitutes a vehicle, cyclist, or pedestrian.

mum number of agents in a scene (as is likely necessary in frameworks like Jax), we only allocate memory equal to the actual number of instantiated agents.

## 3.2 SIMULATOR FEATURES

We provide an overview of some of the pertinent simulator features as well as sharp edges and limitations of the simulator as a guide to potential users.

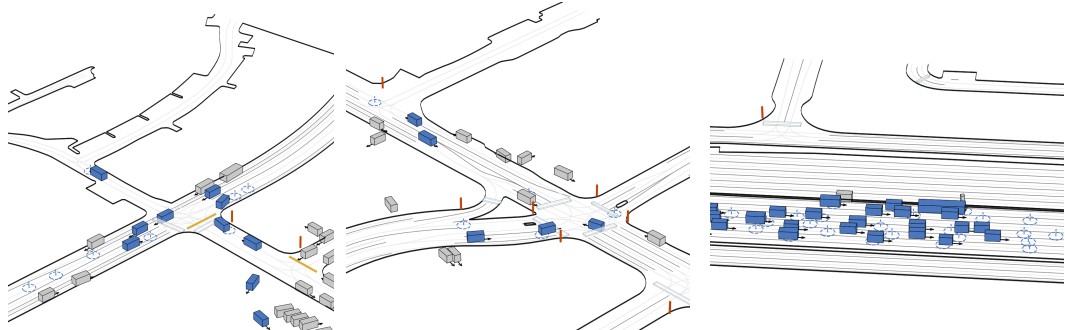

Figure 2: Example scenarios from the Waymo Open Motion Dataset rendered in GPUDrive. The blue boxes and circles indicate agents and their respective destinations.

**Dataset.** GPUDrive represents its map as a series of polylines and does not require a connectivity map of the lanes. As such, it can be made compatible with most driving datasets given the pre-processing of the roads into the polyline format. Currently, GPUDrive supports the Waymo Open Motion Dataset (WOMD) (Ettinger et al., 2021) which is available under a non-commercial license. We show four representative example scenarios from the WOMD in Figure 2. The WOMD consists of a set of over 100,000 multi-agent traffic scenarios, each of which contains the following key elements: 1) Road map - the layout and structure of a road, such as a highway or parking garage. 2) Logged human trajectories from vehicles, cyclists, and pedestrians. 3) Road objects, such as stop signs and crosswalks. Figure 7 depicts an example of an intersection traffic scenario as rendered in GPUDrive.

**Sensor modalities.** GPUDrive supports a variety of observation spaces intended to enable heterogeneous types of agents. Fig. 7 depicts the three types of supported state spaces. The first mode is somewhat unphysical in which all agents and road objects within a fixed radius are observable to the agent. This mode is intended primarily for debugging and quick testing, enabling a user to minimize the amount of partial observability in the environment. The other two modes are based on a GPU-accelerated LIDAR scan, representing what an autonomous vehicle would be able to see and what a human would likely be able to see respectively. Both modes are based on casting LIDAR rays; to model human vision we simply restrict the LIDAR rays to emanate in a smaller, controllable-sized cone that can be rotated through an action corresponding to head rotation. Note that since all objects are represented as bounding boxes of fixed height, the LIDAR observations are over-conservative as, in reality, it is frequently possible to see over the hoods of cars as their height is lower than the body of the rest of the car.

**Agent dynamics.** By default, agents are stepped using a standard Ackermann bicycle model (Details in Appendix B) with actions corresponding to steering and acceleration. This model enables the dynamics of objects to be affected by their length, creating different dynamics for small cars as opposed to large cars like trucks. However, this model is not fully invertible which can make it challenging to use as a model for imitation learning. To enable full invertibility for imitation learning, we also support the simplified bicycle model, taken from Waymax (Gulino et al., 2024), which is a double-integrator in the position and velocity and updates its yaw as:

$$\theta_{t+1} = \theta + s_t(v_t \Delta t + \frac{1}{2} a_t \Delta t^2)$$

where $\theta$ is the yaw, $s$ is the steering command, $v$ is the velocity, and $a$ is the acceleration at time $t$ respectively. $\Delta t$ is the timestep. This model is always invertible given an unbounded set of steering and acceleration actions but is independent of the vehicle length. See the appendix for full details on the models.

Note that this model does not factor in the length of the car, causing both long and short objects to have identical dynamics. However, we have observed that computing the expert actions and then using them to mimic the expert trajectory under this model leads to lower tracking error than the default bicycle model.

**Rewards.** All agents are given a target goal to reach; this goal is selected by taking the last recorded position in the vehicle's logged trajectory. A goal is reached when agents are within some configurable distance $\delta$ of the goal. By default, agents in GPUDrive receive a reward of $1$ for achieving their goal and otherwise receive a reward of $0$. There are additional configurable collision penalties or other rewards based on agent-vehicle distances or agent-road distances though these are not used in the experiments reported in this work.

**Termination conditions.** We terminate an agent's episode when they achieve their goal position and support an option to also terminate the episode if the agent collides. As it is not clear where an agent should go next after it reaches its goal, we simply remove it from the scene afterwards (we note that an alternative might be to generate a new goal for the agent to drive to). Car and cyclist agents are considered to be in collision when they drive through a road edge while pedestrian agents are allowed to cross road edges.

**Environment interface.** As GPUDrive is implemented in C++, we provide a Pythonic interface through nanobind (Jakob, 2022). We create environments for both `torch` and `jax` that conform to the Gymnasium API (Towers et al., 2024) so users can use the simulator entirely through Python if they prefer.

**Available driving simulation agents.** We use reinforcement learning to train a set of agents that reach their goals 95% of the time on a subset of 1000 training scenes. While this number is far below the capability of human drivers, these agents are reactive in a distinct fashion from parametrized driver models in other simulators. In particular, many logged-data simulators construct reactivity by having the driver follow along its logged trajectory but decelerate if an agent passes in front of it. In contrast, these agents can maneuver and negotiate without remaining constrained to a logged trajectory. These trained agents are extremely aggressive about reaching their goals and can be used as an out-of-distribution test for proposed driving agents. The training procedure and more details can be found in Section 4.2.

**Simulator sharp-edges.** We note the following limitations of the benchmark:

- *Absence of a map.* The current version of the simulator does not have a well-defined notion of lanes or a higher-level road map which makes it challenging for algorithmic approaches that require maps. The absence of this feature also makes it challenging to define rewards such as "stay lane-centered."

- *Convex objects only.* Collision checking relies on the objects being represented as convex objects.

- *Unsolvable goals.* Due to incorrect labels of some road points in the Waymo dataset, such as an exit to a parking lot being labeled as an impassable road edge, some agent goals (roughly 2%) are unreachable. For these agents, we default them to simply replaying their logged trajectory and do not treat them as agents.

- *Initialization modes*: In many scenarios, a significant portion of agents (25-75%) are already at or very close to their goal positions, such as parked cars. To highlight the difficulty of controlling agents, GPUDrive supports different initialization modes. By default, the initialization mode is set to "all nontrivial" meaning that only agents more than 2 meters away from their target positions are considered controllable. Note that while this sharply reduces the number of agents in the scene, it more accurately represents actual driving challenges.

## 4 SIMULATOR PERFORMANCE

The following Sections describe the simulator speed. Section 4.1 first shows the raw simulator speed and peak goodput (throughput achieved by the valid number of agents in a scene). Section 4.2 then investigates the impact on reinforcement learning workflows by evaluating the time it takes to train reinforcement learning agents through Independent PPO (IPPO) (Yu et al., 2022), a widely used multi-agent learning algorithm. The performance results here are from IPPO implemented with `PufferLib` (Suarez, 2024). A slower IPPO implementation based on Stable Baselines 3 (Hill et al., 2018) is also available in the repo.

### 4.1 SIMULATION SPEED

Since scenarios contain a variable number of agents, we introduce a metric called Agent Steps Per Second (ASPS) to measure the sample throughput of the simulator. We define the ASPS as the total number of agents across all worlds in a batch that can be fully stepped in a second:

$$\text{ASPS} = \frac{S \times \sum_{k=1}^{N} |A_k|}{\Delta T} \tag{1}$$

where $A_k$ is the set of agents in the $k^{th}$ world, $S$ is the number of steps taken, and $\Delta T$ is the number of seconds elapsed. Figure 3 examines the scaling of the simulator as the number of simulated worlds, which represents the amount of parallelism, increases. To measure performance, we sample random batches of scenarios of size equal to the number of worlds, so that every world is a unique scenario with $K$ agents. On the left-hand side of Figure 3, we compare the performance of GPUDrive to Nocturne (Vinitsky et al., 2022) (CPU, no parallelism), a CPU-accelerated version of Nocturne via Pufferlib (16 CPU cores) (Suarez, 2024) and Waymax Gulino et al. (2024). Empirically, the maximum achievable ASPS of Nocturne is 15,000 (blue dotted line) though we note that additional speedups may be possible. **GPUDrive achieves a peak ASPS of 2.3 million steps**, which is 2 to 3 orders of magnitude faster compared to Nocturne. This performance also surpasses that of Waymax (Gulino et al., 2024), a JAX-based simulator, where we could not run more than 16 environments in parallel due to Out of Memory (OOM) issues.

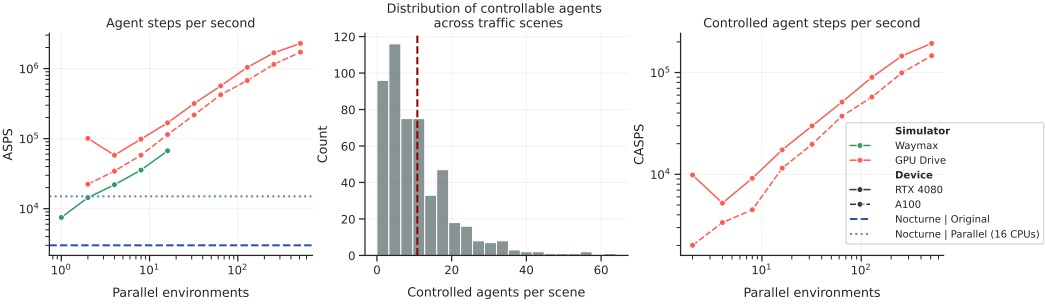

Figure 3: **Peak goodput of GPUDrive on a consumer-grade and datacenter-class GPU compared to original, CPU and GPU-based, implementations using the radial filter observation**. *Left*: The total number of agent steps per second (ASPS) is the number of objects for which our system computes observations at each time step. To ensure a fair comparison, we align the conditions with those used in (Gulino et al., 2024), where all cars, bicyclists, and pedestrians are considered valid experience-generating agents. *Center*: The distribution of controllable agents across 512 scenarios in the Waymo Open Motion Dataset $\mu \approx 10.8$ (red line), $\sigma \approx 9.3$). These numbers are obtained using the "nontrivial" initialization mode in GPUDrive, which initializes only the agents that are more than 2 meters away from their final position. *Right*: The total number of controllable agent steps per second (CASPS) as we increase the number of worlds (parallelism).

In addition to the ASPS, we report another metric to indicate the number of *controllable* agents that are stepped per second. In the Waymo Open Motion Dataset, each scenario contains a varying number of moving agents (Examples in Figure 2). By default, our system only classifies something

as a controllable agent if its movement is necessary to achieve the goal. Therefore, parked cars throughout the episode are not considered controllable agents. To illustrate what this looks like, we plot the distribution of controllable agents across a subset of scenes in Figure 3.

Considering the variance of controllable agents per scene, the right-hand side of Figure 3 depicts the Controlled Agent Steps Per Second (CASPS). CASPS reflects the expected performance of our system when utilizing the Waymo Open Motion Dataset with a *randomly sampled* subset of scenarios. Due to the significant variability in agents throughout the dataset, the highest CASPS is notably lower than the ASPS, at around 200,000. Note that this can be improved by strategically selecting scenes, such as filtering for dense scenes with a high number of controllable agents.

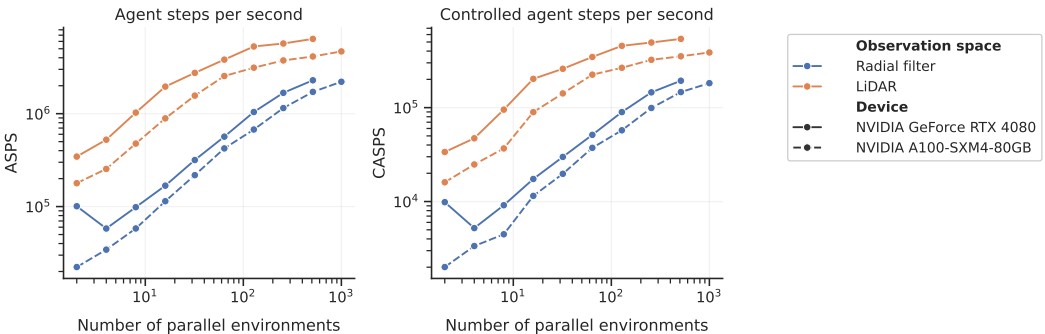

Figure 4: **ASPS and CASPS comparison between Radial Filter and LiDAR observation types**. The radial filter is slower than lidar due to its linear scan of nearby objects. In contrast, the LiDAR observation type is GPU-accelerated, delivering significantly enhanced performance. As demonstrated in the plots, depending on the scene selection, the LiDAR can achieve a speedup of approximately 3x over the radial filter.

Lastly, we demonstrate the speed of different observation spaces in Figure 4. We observe that using LiDAR is approximately three times faster than using the radial filter observation (Details about the supported observation spaces are found in Section 3.2).

## 4.2 END-TO-END SPEED AND PERFORMANCE

The purpose of GPUDrive is to facilitate research and development in multi-agent algorithms by 1) reducing the completion time of experiments, and 2) enabling academic research labs to achieve scale on a limited computing budget. Ultimately, we are interested in the rate at which a machine learning researcher or practitioner can iterate on ideas using GPUDrive. This section highlights what our simulator enables in this regard by studying the end-to-end process of learning policies in our simulator.

Figure 5 contrasts the number of steps (experience) and the corresponding time required to *solve* 10 scenarios from the WOMD between Nocturne (Vinitsky et al., 2022) and GPUDrive. We compare Nocturne to GPUDrive, as it is the closest simulator in functionality, except that it is implemented on the CPU. For benchmarking purposes, we mark a scene as solved when agents can navigate to their designated target position $95\%$ of the time without colliding or going off-road. Ceteris paribus (Details in Appendix D), GPUDrive achieves a 200 - 300x training speedup, solving 10 scenarios in less than 3 minutes compared to approximately 10 hours in Nocturne.

In absolute terms, the number of controlled agent steps per second for the Pufferlib PPO implementation ranges from 200K to 500K, depending on the hardware used and factors such as the number of visible road points per agent.

As shown in Fig. 5, GPUDrive allows us to solve scenes in minutes. Next, we investigate how the *individual scene completion time*, the time it takes to solve a single scenario, changes as we increase the total number of scenarios we train on. In practice, it may be desirable to train agents on thousands of scenarios. Therefore, we ask whether it is feasible to fully leverage the simulator's capabilities with a single GPU.

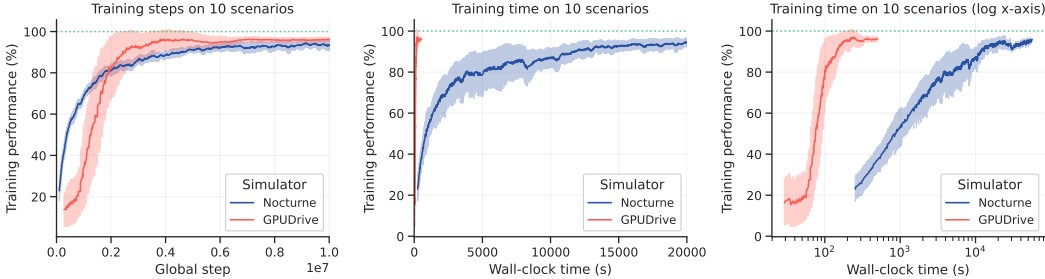

Figure 5: **From hours to seconds.** *Left*: Training performance (goal-reaching rate) as a function of the global step of the controlled agents (CASPS). *Center*: Training performance as a function of wall-clock time in seconds. *Right*: Training performance as a function of wall-clock time where the x-axis is on a *log scale*. Runs are averaged across three seeds, replicating environmental and experimental conditions as closely as possible. See Appendix D for the hyperparameters and training details. The green dotted line marks optimal performance (all agents reach their goal without collisions).

Interestingly, we find that the amortized sample efficiency increases with the size dataset of scenes we train in. Figure 6 shows the average completion time per scenario as we increase the dataset. For instance, using IPPO with 32 scenarios takes 2 minutes per scenario. In contrast, solving 1024 unique scenarios takes about 200 minutes, which amounts to only 15 seconds per scenario. We expect that these scaling benefits will continue as we further increase the size of the training dataset. This suggests that GPUDrive should enable effective utilization of the large WOMD dataset comprising 100,000 diverse traffic scenarios, even with limited computational resources.

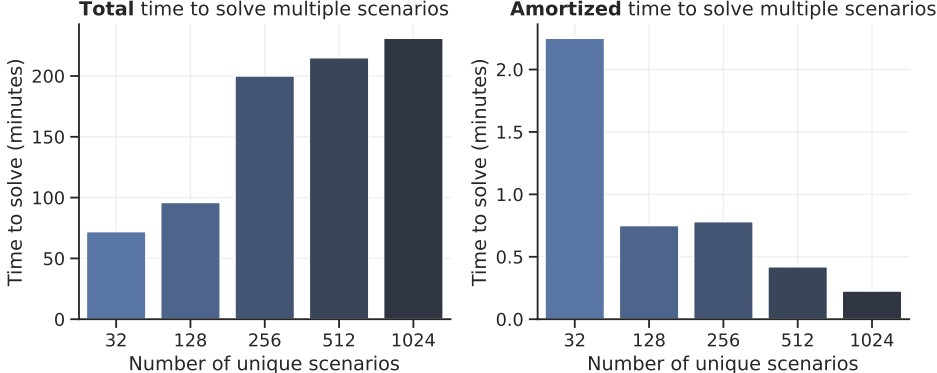

Figure 6: **Scale reduces individual scene completion time**. *Left*: Total time required to solve a fixed number of scenarios to a goal-reaching rate of 95%. Note that time-to-completion is sub-linear concerning the number of scenes. *Right*: Each additional scenario costs less to solve than the previous scenario. At 1024 scenes, the per-scene cost of solving an additional scene is on the order of 15 seconds.

### 4.3 MEASURING REMAINING UNSOLVED TASKS

The Waymo Open Motion Dataset contains a total of 103,354 traffic scenarios (Ettinger et al., 2021). In this paper, we demonstrate that an agent can achieve 95% performance on a subset of 1000 scenes after 15 hours of training. Additionally, we show that the time required to solve scenarios decreases as the dataset size increases. Our analysis of the failure rates in the current best-performing policy suggests that the goal-reaching limit is around 98%, due to mislabeled road edges in the dataset, which render some goals unreachable because they are located beyond uncrossable roads. An important direction for future work is developing agents that achieve near-perfect performance,

with a goal-reaching rate approaching 100% and a 0% collision rate while addressing the limitations posed by data inaccuracies in the benchmark.

## 5 CONCLUSION

In this work, we present GPUDrive, a GPU-accelerated, multi-agent, and data-driven simulator. GPUDrive is intended to help generate the billions of samples that are likely needed to achieve effective reinforcement learning for multi-agent driving planners. By building atop the Madrona Engine (Shacklett et al., 2023), we can scale GPUDrive to hundreds of worlds with potentially thousands of agents leading to throughput of millions of steps per second. This throughput occurs while synthesizing complex observations such as LiDAR. We show that this throughput has consequent implications for training reinforcement learning agents, leading to the ability to train agents to solve any particular scene in minutes and in seconds when amortized across many scenes. We release the simulator and integrated trained agents to enable further research.

**Future work and simulator extensions.**    This paper represents an initial step toward scaling reinforcement learning for multi-agent planning in safety-critical, mixed human-autonomous settings. Several important opportunities remain for future work:

- Agent performance: Training agents to navigate without crashing in any scenario, matching human capabilities, remains an unresolved challenge. Currently, agents trained in GPU-Drive achieve a 95% success rate in a few hours, but this still falls short of the standards we aim for. Getting to 100% goal reaching performance is left for future work.

- Integrating multiple datasets: Future work will focus on integrating various datasets, such as NuPlan or NuScenes (Caesar et al., 2020). This integration will enable training on data from multiple geographic locations, including Boston and Singapore. To achieve this, we need to establish a data-processing pipeline that ensures the datasets are converted into a format compatible with the simulator.

- Diverse realistic sim agents: We plan to extend GPUDrive by introducing a variety of simulation agents that represent a broad range of human-like behaviors. This will improve sim realism.

## 6 REPRODUCIBILITY STATEMENT

The simulator's source code and data are available at `https://github.com/Emerge-Lab/gpudrive`. To ease reproducibility, we provide Dockerfiles to simplify setup. The experiments in the paper can be reproduced using a single file on a single A100 in 16 hours or slightly longer for less performant hardware. The data has been pulled from the Waymo Motion dataset and parsed into JSON files that are used to initialize the simulator; these are all available from Huggingface datasets.

## 7 ETHICS STATEMENT

This paper presents GPUDrive, a GPU-accelerated simulator for multi-agent learning in autonomous driving. We use publicly available datasets, such as the Waymo Open Motion Dataset (Ettinger et al., 2021), which are anonymized to protect privacy. GPUDrive is intended for research purposes and not for real-world deployment without further validation. We recognize the risks of autonomous systems and emphasize the importance of safety and fairness in their application. Although this work does not involve human participants directly, it leverages real-world data that may reflect human behavior, and we strive to avoid harm or discrimination. The codebase, pre-trained agents, and evaluation scripts are openly available to promote transparency. There are no conflicts of interest or external sponsorships affecting this research. We are committed to ethical practices in data handling, model deployment, and research integrity.

ACKNOWLEDGMENTS

This work is funded by the C2SMARTER Center through a grant from the U.S. DOT's University Transportation Center Program. The contents of this report reflect the views of the authors, who are responsible for the facts and the accuracy of the information presented herein. The U.S. Government assumes no liability for the contents or use thereof. This work was also supported in part through the NYU IT High-Performance Computing resources, services, and staff expertise.

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

## A    REPRODUCIBILITY

### A.1    EXTRA FIGURES

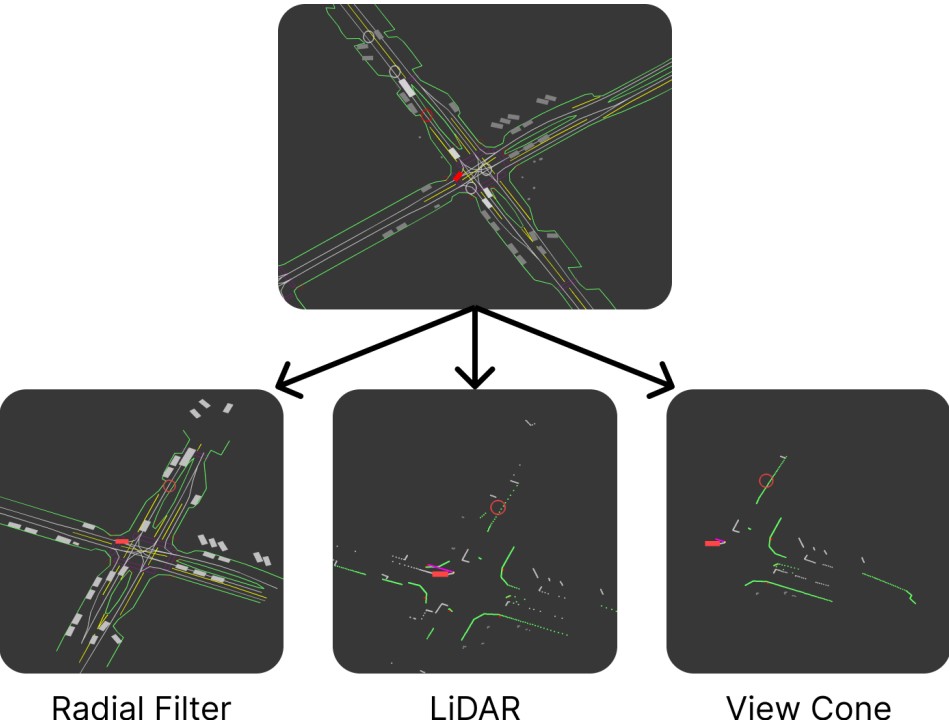

Figure 7: Visualization of different observation spaces available in GPUDrive. The top scene is an example scenario from the dataset, rendered from the ego-centric perspective of the red vehicle. Grey cars are parked cars while white cars are other controlled agents. From left to right: the Radius Filter returns all objects within 100 meters, the LIDAR observation with 3000 rays spread around 360 degrees, and a view cone consisting of 3000 rays emanating in a 120-degree view cone.

### A.2    CODE REPRODUCIBILITY

All code required to reproduce the paper is open-sourced at https://github.com/ Emerge-Lab/gpudrive/tree/main under release number v0.4.0

### A.3    COMPUTATIONAL RESOURCES

All RL experiments in this paper were run on an NVIDIA RTX 4080 or A100. Total resources for the paper correspond to less than 2 GPU days.

## B    DYNAMICS MODEL

Agents are driven by a kinematic bicycle model (Rajamani, 2011) which uses the center of gravity as reference point. The dynamics are as follows. Here $(x_t, y_t)$ stands for the coordinate of the vehicle's position at time $t$, $\theta_t$ stands for the vehicle's heading at time $t$, $v_t$ stands for the vehicle's speed at time $t$, $a$ stands for the vehicle's acceleration and $\delta$ stands for the vehicle's steering angle. $L$ is the distance from the front axle to the rear axle (in this case, just the length of the car) and $l_r$ is the distance from the center of gravity to the rear axle. Here we assume $l_r = 0.5L$.

$$\dot{v} = a$$
$$\bar{v} = \text{clip}(v_t + 0.5 \; \dot{v} \; \Delta t, -v_{\text{max}}, v_{\text{max}})$$
$$\beta = \tan^{-1}\left(\frac{l_r \; \tan(\delta)}{L}\right)$$
$$= \tan^{-1}(0.5 \; \tan(\delta))$$
$$\dot{x} = \bar{v} \; cos(\theta + \beta)$$
$$\dot{y} = \bar{v} \; sin(\theta + \beta)$$
$$\dot{\theta} = \frac{\bar{v} \; \cos(\beta) \; \tan(\delta)}{L}$$

We then step the dynamics as follows:

$$x_{t+1} = x_t + \dot{x} \; \Delta t$$
$$y_{t+1} = y_t + \dot{y} \; \Delta t$$
$$\theta_{t+1} = \theta_t + \dot{\theta} \; \Delta t$$
$$v_{t+1} = \text{clip}(v_t + \dot{v} \; \Delta t, -v_{\text{max}}, v_{\text{max}})$$

## C  LICENSE DETAILS AND ACCESSIBILITY

Our code is released under an MIT License. The Waymo Motion dataset is released under a Apache License 2.0. The code is available at https://github.com/Emerge-Lab/gpudrive.

# D  TRAINING DETAILS

## D.1  END-TO-END PERFORMANCE

The Table below depicts the hyperparameters used to produce the results in Section 4.2.

Table 2: Experiment hyperparameters used for comparing the training runs between Nocturne and GPUDrive in Figure 5. The environment configurations are aligned as closely as possible, using the same observations and field of view. The dataset includes the same 10 scenarios. It's important to note that the length of the GPUDrive rollout is approximately equal to the number of worlds multiplied by the rollout length and then multiplied by the number of controllable agents. We have set this value to be $92 \times 50 \approx 4600$ to approximately match the rollout length in Nocturne.

| Parameter | IPPO `GPUDrive` | IPPO `Nocturne` |
|---|---:|---:|
| $\gamma$ | 0.99 | 0.99 |
| $\lambda_{\text{GAE}}$ | 0.95 | 0.95 |
| PPO rollout length | 92 | 4096 |
| PPO epochs | 5 | 5 |
| PPO mini-batch size | 2048 | 2048 |
| PPO clip range | 0.2 | 0.2 |
| Adam learning rate | 3e-4 | 3e-4 |
| Adam $\epsilon$ | 1e-5 | 1e-5 |
| normalize advantage | yes | yes |
| entropy bonus coefficient | 0.001 | 0.001 |
| value loss coefficient | 0.5 | 0.5 |
| seeds | 42, 12, 67 | 42, 12, 67 |
| number of worlds | 50 | 1 |

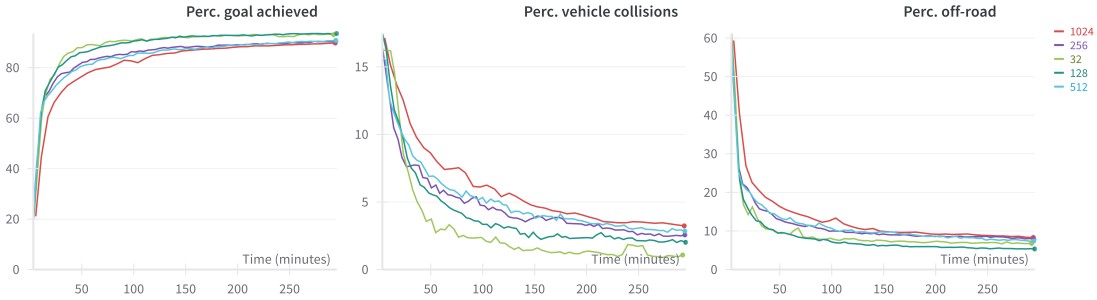

Figure 8: **Key performance metrics as a function of training time grouped by the number of unique scenes in a batch reported in Figure 6.** *Left*: The aggregate percentage of agents that achieved their goal. *Center*: The aggregate percentage of agents that collided with another vehicle. *Right*: The aggregate number of vehicles that crossed a road edge.

