# OpenReview forum: "GPUDrive: Data-driven, multi-agent driving simulation at 1 million FPS"
_ICLR.cc/2025/Conference — ICLR 2025 Poster_

### Official Review · Reviewer_dLhs · 2024-10-27

**Soundness:** 3
**Presentation:** 2
**Contribution:** 2
**Rating:** 3
**Confidence:** 4

**Summary:**

This paper presents GPUDrive, a GPU-based simulator for autonomous driving. The simulator is compatible with existing datasets and allows parallel simulations. The experiments show that it can train a policy with 25-40x training speedup against the baseline.

**Strengths:**

1. The GPU-based simulation is important in facilitating the training for complex real-world applications, highlighted by recent advances in robotics.
2. The experiment result shows significant wall-clock time speedup again baselines.

**Weaknesses:**

1. The paper claims compatibility with existing datasets but only demonstrates map loading, leaving other functionalities unclear. For example, the imitation learning experiment or mixing agent behaviors—some from datasets and others from RL agents during training.
2. I believe one major advantage of parallel environments is that it allows you to do randomization across different environments (worlds). However, the paper lacks detail on whether GPUDrive supports this capability.
3. While GPUDrive offers a Python interface, I am curious how easy to customize those key elements in the environment given that the observation, reward, dynamic functions are written in C++.
4. Experiments only evaluate IPPO, despite the paper claims that it targets at mixed motive setting.

**Questions:**

1. In Figure 3, the speedup appears nearly linear. However, it would be helpful to examine scaling performance by adding more environments to identify saturation points and gain insights into system limitations..
2. What is the scaling of speedup with respect to the number of agents in the environment? e.g., fix the number of environments and scales the number of agents?
3. In Figure 5, do you use the CPU-parallel version of Nocturne?

---

> ### Author Response · Authors · 2024-11-15
> **Response to reviewer dLhs**
>
> Thank you for recognizing the _significant speedup_ provided by our simulator and its potential to enable research in complex real-world scenarios. Below are our clarifications:
>
> ### Sim functionality and mixing agent behaviors
>
> > _The paper claims compatibility with existing datasets but only demonstrates map loading, leaving other functionalities unclear. For example, the imitation learning experiment or mixing agent behaviors—some from datasets and others from RL agents during training._
>
> We highlight that **GPUDrive fully supports mixing agent behaviors**; you can combine replay agents, scripted agents, pre-trained agents, and RL agents in a single rollout without sacrificing parallelism. $\color{green}{\text{We have added this and plans for future work in this regard to Section 5 of the paper.}}$
>
> ### Domain randomization
>
> > _I believe one major advantage of parallel environments is that it allows you to do randomization across different environments (worlds). However, the paper lacks detail on whether GPUDrive supports this capability._
>
> Domain randomization is certainly a valuable feature for any simulator. While it is already possible to implement some types of domain randomization, such as randomizing goals or initial positions, we believe the primary challenge of driving all agents to the observed goals remains unsolved for now. We plan to explore domain randomization in future work and appreciate the reviewer’s suggestion.
>
> ### Customization
>
> > _While GPUDrive offers a Python interface, I am curious how easy it is to customize those key elements in the environment given that the observation, reward, and dynamic functions are written in C++._
>
> Thank you for pointing this out. We believe GPUDrive is easy to extend despite being in C++ for two key reasons:
>
> 1. We offer **extensive Python bindings that cover most expected observations, reward functions, and dynamics**, allowing users to customize many components without having to touch the C++ code.
> 2. **C++ remains a popular language**, particularly in the robotics and autonomous vehicle communities, and many of our users have the necessary familiarity to extend the framework. Additionally, the framework handles parallelism, so even a basic understanding of C++ is sufficient for implementing new rewards or dynamics.
>
> ### Algorithms
>
> > _Experiments only evaluate IPPO, despite the paper claims that it targets a mixed-motive setting._
>
> The primary goal of the paper is to propose a fast multi-agent simulator, with the RL experiments and code included to help users get started. We note that **IPPO is a valid solver for mixed games and performs well empirically**. In many multi-agent problems, it has been observed that PPO is an effective solver (See e.g., MAPPO [1]).
>
> If the reviewer is asking whether the "independent" aspect of IPPO is compatible with a general-sum game, the answer is yes—independence refers solely to decentralized training.
>
> ### Questions
>
> > _In Figure 3, the speedup appears nearly linear. However, it would be helpful to examine scaling performance by adding more environments to identify saturation points and gain insights into system limitations._
>
> We apologize for the confusion, please note that **this is a log-log plot**. The plot shows that as we increase the number of worlds (x) the throughput increases at the same rate (y).
>
> > _In Figure 5, do you use the CPU-parallel version of Nocturne?_
>
> Yes! For a fair comparison, we plot both the single-CPU (striped blue) and the parallelized Nocturne using PufferLib (dotted blue, 16 CPUs). Note that the PufferLib version has been carefully designed to outperform naive Python multiprocessing.
>
> ### Conclusion
>
> Thank you for your valuable feedback! We hope we have addressed the reviewer's comments and are happy to further discuss any of these points. We kindly ask that if we have addressed the reviewer's concerns, they consider increasing their support for our paper.
>
> ### References
>
> **[1]** Yu, C., Velu, A., Vinitsky, E., Gao, J., Wang, Y., Bayen, A., & Wu, Y. (2022). The surprising effectiveness of ppo in cooperative multi-agent games. Advances in Neural Information Processing Systems, 35, 24611-24624.

---

> > ### Comment · Reviewer_dLhs · 2024-11-16
> > **Response to authors**
> >
> > Thank you for your efforts during the discussion phase. After thoroughly reviewing your response, it appears that sim functionality and domain randomization remain part of the development plan and unsupported. For example, there is no experiments to verify the claim of supporting mixed agent behaviors. Domain randomization is also currently missing, which is an important feature in parallel simulator.
> >
> > Additionally, there is no analysis provided on cooperative or competitive behaviors, given that the paper strengths the mixed-motive setting in the Intro. From my perspective, IPPO is a rather basic algorithm and differs significantly from MAPPO. But I note this is a minor weakness.
> >
> > Furthermore, I find the statement, "The plot shows that as we increase the number of worlds (x), the throughput increases at the same rate (y)," to be unclear. Does this imply that the simulator has no limit on parallelization? If there is a limit (assume you use commercial GPUs), linear scaling would eventually plateau as the number of environments increases.
> >
> > I appreciate the authors’ efforts in building this simulator. However, some claims appear to be overstated and unsupported by experimental results. As an engineering paper, I think it is not ready from being accepted and will maintain my score.

---

> ### Author Response · Authors · 2024-11-17
> **Hopefully helpful clarification**
>
> Hi,
>
> We appreciate the rebuttal and discussion. We have attempted to clarify some of the points below.
> If we can restate your objections to the paper they appear to be the following:
>
> - **The simulator is written in C++.**
> We assume since this was not brought up in the response that we are in agreement that this is not an issue since most simulators are actually written in C++ or other languages and just have python bindings.
> - **The sim does not currently support domain randomization.**
> While we agree that domain randomization would be an interesting feature to have, we are unclear why it is a critical feature. The simulator already comes with 450000 unique scenes and is not readily solved by extensive tuning of an RL algorithm. Is it possible that there was confusion and the existence of 450000 unique scenes was not clear? If so, we appreciate that being pointed out and will rewrite it.
> - **Using IPPO instead of MAPPO.**
> Note that our usage of an RL algorithm is not to make claims about algorithms but purely to point out how quickly scenes can be solved in the benchmark. If it would lead the reviewer to increase their score, we can run MAPPO. We are familiar with the distinction between IPPO and MAPPO and it is unclear to us why MAPPO would make a difference here since the empirical differences between the two are small. For example, in the MAPPO paper, centralized value functions only really helped in 3 and 4 player Hanabi.
> - **The simulator does not provably contain mixed games.**
> We want to caution that there are two possible uses of the word mixed and we are not sure which one the reviewer is referring to. In the first case, mixed agents i.e. multiple policies, the simulator does support this and we have updated the text already to clarify it. If the reviewer is referring to the simulator containing general sum games, whether it does or does not is a function of what reward functions the user chooses to use. Under the default reward of goal reaching, the game is general sum and also contains conflicts of interest since this is a standard occurrence in driving (for example, at intersections).
> - **"The plot shows that as we increase the number of worlds (x), the throughput increases at the same rate (y)," to be unclear. Does this imply that the simulator has no limit on parallelization? If there is a limit (assume you use commercial GPUs), linear scaling would eventually plateau as the number of environments increases.**
> Our point, which we did not make clearly, was that the simulator experiences sublinear scaling but speed is still improving with more environments. We cannot run more than 1000 or so environments as we exhaust all the GPU memory at that point.
>
> Hopefully, those points clear up some of the issues?
>
> **Finally, we want to re-address the point about user difficulty in programming the simulator.** We claim a mono-language implementation in Python is neither sufficient nor necessary to meet our dual goals of productivity and performance.
> From a language perspective, the GPUDrive architecture can be decomposed as follows:
> 1. A learning layer (Python)
> 2. A bridging layer (C++)
> 3. CUDA kernels.
>
> Why is it hard to do it any other way?
> 1. CUDA kernels must be written in CUDA-C++ because NVIDIA does not support JIT compilation for any language other than C++, and JIT compilation is upstream of maximizing performance on NVIDIA GPUs. For instance, by opting for JIT compilation instead of the more classic Ahead-Of-Time compilation we enable “Runtime LTO”. We refer the reviewer to [JIT LTO](https://docs.nvidia.com/cuda/cufft/ltoea/usage/jit_lto.html) for an explanation of the performance benefits of this feature.
> 2. Though written in C++, we argue the Bridging layer does not decrease productivity. A simplified view of the PyTorch architecture is that it exports Python bindings to CUDA kernels. Just as with GPUDrive's architecture, the Python bindings are bridged to CUDA via C++. We observe this makes PyTorch no less productive for end-users.
>
> We could of course attempt to rewrite the simulator using an array-based programming language like PyTorch or Jax. However, implementing complex training environments using state-of-the-art simulation methods requires complex data structures and non-trivial control flow (traversing acceleration structures, collision solvers, state machines, conditional logic in functions) is cumbersome in array-based abstractions. For this reason, Jax and Torch-based environments rarely contain all these features.

---

### Official Review · Reviewer_geDV · 2024-11-02

**Soundness:** 3
**Presentation:** 3
**Contribution:** 3
**Rating:** 8
**Confidence:** 4

**Summary:**

This paper proposes GPUDrive, a GPU-accelerated multi-agent driving simulator designed to increase efficiency of learning-based systems. The simulator allows for loading expert trajectories from real-world driving datasets, can support multiple observation spaces (including e.g. LiDAR), and displays favorable throughput compared to other openly accessible simulators. The simulator, together with pre-trained goal-conditioned policies, is made openly available with accessible pythonic interfaces.

**Strengths:**

- The proposed simulator improves over alternatives in terms of sample efficiency. All of the design choices appear reasonable, while the underlying source code with pre-trained driving baselines will be released.
- The ability to load real-world driving datasets is extremely useful, while providing a variety of observation spaces is a great feature.
- Transparency about current limitation of the benchmark are very helpful for user adaptation.

**Weaknesses:**

- While the focus of this paper is on providing a novel simulator, it would be very interesting to see some more complex behavior over longer time-horizons to fully capture the capabilities unlocked by the simulator (e.g. training a single agent policy with higher velocity limit to weave through a simulated traffic scene, etc.)
- Showcasing such behavior would likely require addressing the “Absence of a map” limitation raised in the paper, in order to formulate more sophisticated reward function. An important question would then be how easily this could be integrated, and how much the absence of such a feature could hurt adaptation of the simulator.
- The discussion of batched simulators could be extended to include references [1-3], where [1] has driven many results in single-agent robot learning, while [3] considers heterogenous multi-agent settings
- Figure 3 mentions performance on an RTX 4080, while line 711 states RTX 8000
- Line 308: “number valid number”

**References**

[1] V. Makoviychuk, L. Wawrzyniak, Y, Guo, M. Lu, K. Storey, M. Macklin, D. Hoeller, N. Rudin, A. Allshire, A. Handa, and G. State. “Isaac gym: High performance gpu-based physics simulation for robot learning.” NeurIPS, 2021.

[2] J. Panerati, H. Zheng, S. Zhou, J. Xu, A. Prorok, and A. P. Schoellig. "Learning to fly—a gym environment with pybullet physics for reinforcement learning of multi-agent quadcopter control.” IROS, 2021.

[3] M. Lechner, L. Yin, T. Seyde, T.-H. Johnson Wang, W. Xiao, R. Hasani, J. Rountree, and D. Rus. “Gigastep - one billion steps per second multi-agent reinforcement learning." NeurIPS, 2024.

**Questions:**

- How easily can the simulator be updated to efficiently provide map-like utilities that allow for lane-keeping rewards (re mentioned limitations)?
- Do you support loading multiple different polices for individual agents? Could they have different sampling rates? How would these aspects affect efficiency?
- How do traffic jams affect throughput (re BVH)? This could be an interesting experiment to add.
- In video scene_53.mp4, agent 4 displays rather jerky behavior when moving towards its goal - could you elaborate on the underlying reasons?
- In video scene_43.mp4, agents 1 and 10 seemingly disappear without reaching their goals - could you elaborate on this behavior?

---

> ### Author Response · Authors · 2024-11-15
> **Response to reviewer geDV**
>
> Dear reviewer
>
> Thank you for your thoughtful comments and for noticing GPUDrive is built on top of a real-world driving dataset (WOMD), which enhances sim realism. Please find our responses below:
>
> ### Complex behaviors
>
> > _While the focus of this paper is on providing a novel simulator, it would be very interesting to see some more complex behavior over longer time-horizons to fully capture the capabilities unlocked by the simulator (e.g., training a single agent policy with higher velocity limit to weave through a simulated traffic scene, etc.)_
>
> We think this is a great suggestion and an interesting direction for future work. As mentioned, the key contribution of the paper is the simulator itself; we look forward to users or future papers showing these behaviors!
>
> > _... showcasing such behavior would likely require addressing the “Absence of a map” limitation raised in the paper, in order to formulate a more sophisticated reward function. An important question would then be how easily this could be integrated, and how much the absence of such a feature could hurt adaptation of the simulator._
>
> The reviewer correctly points out that some features are challenging to implement without a map. While the current simulator task of reaching goals without collisions can be achieved without an explicit map, more complex scenarios would benefit from one.
>
> That said, **we do incorporate map-like features, such as lane lines**, which assist with imitation learning and allow for reward functions like staying lane-centered. Full map connectivity is available in the dataset, and we are actively working to $\color{orange}{\text{integrate full map connectivity into the simulator}}$.
>
> ### References
>
> > _The discussion of batched simulators could be extended to include references [1-3], where [1] has driven many results in single-agent robot learning, while [3] considers heterogeneous multi-agent settings_
>
> Thank you for pointing this out! $\color{green}{\text{We integrated these references in the discussion of batched simulators.}}$
>
> ### Typos
>
> Thank you for catching these!
> > _Figure 3 mentions performance on an RTX 4080, while line 711 states RTX 8000_
>
> $\color{green}{\text{We fixed that typo, it should be RTX 4080}}$.
>
> > _Line 308: “number valid number”_
>
> $\color{green}{\text{Noted and fixed.}}$
>
> ### Questions
>
> > _How easily can the simulator be updated to efficiently provide map-like utilities that allow for lane-keeping rewards (re mentioned limitations)?_
>
> Some required features are already present in the data structure; they can be implemented by adding a reward for staying near lane lines. However, for a more robust implementation of this reward, it would likely be necessary to support maps.
>
> > _Do you support loading multiple different policies for individual agents? Could they have different sampling rates? How would these aspects affect efficiency?_
>
> We do support loading multiple policies per agent, and they could have different sampling rates as long as they are integer multiples of the simulator time-step. Since agents not taking action at a step would simply be stepped, this would only increase the speed of the simulator step.
>
> > _How do traffic jams affect throughput (re BVH)? This could be an interesting experiment to add._
>
> Thank you for the suggestion! We currently do not have an answer, but we agree that studying the impact of traffic jams on throughput (in relation to BVH) would be an interesting avenue for future research.
>
> > _In video scene_53.mp4, agent 4 displays rather jerky behavior when moving towards its goal - could you elaborate on the underlying reasons?_
>
> Great observation and thank you for checking out the videos! These agents are trained solely with a goal-reaching reward; they have no incentive _not_ to drive jerkily. If we added jerk penalties (which can easily be done), this would probably disappear.
>
> > In video scene_43.mp4, agents 1 and 10 seemingly disappear without reaching their goals - could you elaborate on this behavior?
>
> Yes, we configured the simulator to remove vehicles from the scene when they collide with a road edge or another agent, although this behavior can be easily adjusted in the config file. The video shows that the pre-trained policy, which achieved 95% performance over 1000 scenes (colliding 5% of the time), still leaves room for improvement. We are actively working on $\color{orange}{\text{improving both the effectiveness and diversity of the simulation agents}}$.
>
> ### Conclusion
>
> Thank you for taking the time to provide such a thorough review! We believe we have addressed all of your questions and welcome further discussion. If all your concerns have been resolved, we would be thankful if you could consider increasing your support for our paper.

---

### Official Review · Reviewer_Y9u7 · 2024-11-04

**Soundness:** 4
**Presentation:** 3
**Contribution:** 3
**Rating:** 8
**Confidence:** 4

**Summary:**

GPU drive introduces a fast multi agent simulator build using C++ that helps you run complex scenarios especially related to self driving cars at scale built on top of the Waymo Open Motion dataset. This allows iterating on these scenarios quicker reaching greater than a million FPS thus allowing more experimentation runs and iterating/trying out different scenarios even on desktop grade GPU's.

**Strengths:**

1. A multi agent simulator accelerated on the GPU iteration of over a million steps per second.
2. Very well written and structured code to run any experiment easily with a lot of easy experimentation code readily available.
3. Extensive results analyzing the sampling frequency of the simulation.

**Weaknesses:**

1. Figure 2 needs a better caption and an explanation
2. Designed to fit one exact dataset. A section explaining the effort required to integrate other datasets is desirable.

**Questions:**

1. Benchmarks consist of limitations because of the dataset. Can it be addressed by using another dataset ?
2. Stable baselines is not known that well for speed. Could other implementations of PPO have been used ?
3. Is there support for multi GPUs ? And if they do exist, an ablation or benchmark would be great for that

---

> ### Author Response · Authors · 2024-11-15
> **Response to reviewer Y9u7**
>
> Dear Reviewer,
>
> First of all, thank you for your kind words about the _clarity_ and _structure_ of the paper and code. We are excited about the potential of GPUDrive to **enable RL at scale** on a single GPU and make the algorithmic design process _interactive_. Please find our responses to your comments below:
>
> ### Figure 2
> > _Figure 2 needs a better caption and an explanation_
>
> We agree that the caption and explanation could be more detailed. We have $\color{green}{\text{updated the place and caption and added a more comprehensive explanation}}$ in the current version (see Figure 1).
>
> ### Integrating Multiple Datasets
> > Designed to fit one exact dataset. A section explaining the effort required to integrate other datasets is desirable.
>
> We appreciate the suggestion to explain the effort required to integrate other datasets. We are actively working on this and have $\color{green}{\text{added a new paragraph in the current version (see Section 5)}}$ to address this and will point to our data processing code after the rebuttal period.
>
> ### Questions
> - Benchmarks and Dataset Limitations: Yes, we are actively $\color{orange}{\text{integrating additional datasets}}$, such as the Nuscenes (https://www.nuscenes.org/); this simply requires putting the files into our JSON format. Note that most large, diverse datasets have similar limitations as they are collected from the sensors of a single vehicle. While no dataset is perfect, we believe that working with human data, even with its limitations, provides significant value.
> - Stable Baselines and PPO Speed: We acknowledge the limitations of the Stable Baselines 3 (SB3) IPPO implementation. In response, we have implemented an $\color{green}{\text{improved version of IPPO, achieving an end-to-end training throughput of 500K, a 10X speedup}}$ compared to the previous SB3 version.
> - Multi-GPU Support: We do not currently have multi-GPU support as this is a property of the training code and not the simulator. It is straightforward to add multi-GPU support by wrapping the model in torch DDP but we do not mention this in the work. We can include it if it feels important to the reviewer.
>
> ### Conclusion
>
> Thank you for helping us make this work better! We hope we have addressed your comments properly and welcome further discussion.

---

> > ### Comment · Reviewer_Y9u7 · 2024-11-30
> >
> > Thanks for your clarifications and I don't have anything else to add. This paper deserves the 8 I have given :)

---

### Official Review · Reviewer_AUkE · 2024-11-04

**Soundness:** 3
**Presentation:** 3
**Contribution:** 3
**Rating:** 6
**Confidence:** 4

**Summary:**

The paper presents a GPU accelerated simulator that can generate millions of simulation steps samples per second that can be used to train multi-agent reinforcement learning (RL) algorithms. The simulator is claimed to simulate hundreds to thousands of scenarios/scenes in parallel with each scene containing thousands of agents.

The simulator is built on top of the Madrona Game Engine and is written in C++. The C++ simulator engine can also be interfaced with learning environments written in JAX and Torch.
The authors have released implementations of RL algorithms capable of processing millions of agent steps per second and some baseline agents trained on these algorithms that achieve 95% of their goals. The simulator claims to provide both recorded logs and RL agents for the environment.

The authors introduced certain metrics to evaluate the simulation speed of GPUDrive in terms of agent steps per second (ASPS), controllable agent steps per second (CASPS) and scene completion time. Compared against other sim engines like Nocturne GPUDrive achieved 25-40x training speedup solving 10 scenarios in less than 15 minutes.

**Strengths:**

- The proposed simulator has the flexibility to handle multiple modalities of sensor data.

- The authors have implemented ways to reduce the memory footprint due to the large number of agents and observation space using algorithms like Bounding Volume Hierarchy (to exclude certain agent pairs for collision checking) and polyline decimation to approximate the straight polylines.

- The trained agents are claimed to be useful for out-of-distribution tests for the driving agents.

- The authors presented the different simulator features in a comprehensive way.

- The paper shows that the simulator gets the scaling benefits in terms of increased amortized sample efficiency with increasing dataset size. This can be beneficial when dealing with large scale datasets with limited compute.

**Weaknesses:**

- The paper does not provide simple IDM (intelligent driving models) agents that can be sometimes practical to have basic reactivity to the ego-agent.
- The authors mention that the current work is limited in properly utilizing the generated samples for optimal training.

- Just a thought: The implementation is in C++ and it provides a binding interface with Python environments. It would have been nice to have a mono-language (primarily Python based) tool as the model training and other related pipelines are mostly in Python.

**Questions:**

- Were other agents in the scenes like pedestrians and cyclists also controlled? If so, what were the dynamics used to model their behavior if they were not logged?
- Nit: Ethical statement was missing?
- Nit: Can the x-axis in the center plot in Fig 5 be made to a log scale?

---

> ### Author Response · Authors · 2024-11-15
> **Response to reviewer AUkE**
>
> Dear Reviewer,
>
> Thank you for your valuable feedback! We appreciate your recognition of GPUDrive's scaling benefits and mentioning that we present the wide range of simulator features in a _“comprehensible way”_. Please find our response to your comments below:
>
> ### Reactive Sim Agents
> > _The paper does not provide simple IDM (intelligent driving models) agents that can be sometimes practical to have basic reactivity to the ego agent._
>
> We agree that reactivity of simulation agents is crucial. While we do not include simple IDM agents, we **provide pre-trained reactive simulation agents** trained through self-play in the simulator that offer this basic reactivity. The behavior of these agents can be seen in the videos at https://sites.google.com/view/gpudrive/. We are actively working on expanding the $\color{orange}{\text{diversity and effectiveness of available sim agents}}$.
>
> ### End-to-End Training Throughput
> > _The authors mention that the current work is limited in properly utilizing the generated samples for optimal training._
>
> We assume the reviewer is referring to the end-to-end training performance in Section 4.2. We acknowledge the concern regarding sample utilization. We have since addressed this issue and $\color{green}{\text{added an improved IPPO implementation, improving the training speed 10X: from 50K to 500K AFPS. Please see the updated Figure 5 in the paper.}}$
>
> ### Implementation in C++
> > _Just a thought: The implementation is in C++ and it provides a binding interface with Python environments. It would have been nice to have a mono-language (primarily Python based) tool as the model training and other related pipelines are mostly in Python._
>
> We claim a mono-language implementation in Python is neither sufficient nor necessary to meet our dual goals of productivity and performance. From a language perspective, the GPUDrive architecture can be decomposed as follows:
> 1. A learning layer (Python)
> 2. A bridging layer (C++)
> 3. CUDA kernels.
>
> Why is it hard to do it any other way?
> 1. CUDA kernels must be written in CUDA-C++ because NVIDIA does not support JIT compilation for any language other than C++, and JIT compilation is upstream of maximizing performance on NVIDIA GPUs. For instance, by opting for JIT compilation instead of the more classic Ahead-Of-Time compilation we enable “Runtime LTO”. We refer the reviewer to [JIT LTO](https://docs.nvidia.com/cuda/cufft/ltoea/usage/jit_lto.html) for an explanation of the performance benefits of this feature.
> 2. Though written in C++, we argue the Bridging layer does not decrease productivity. A simplified view of the PyTorch architecture is that it exports Python bindings to CUDA kernels. Just as with GPUDrive's architecture, the Python bindings are bridged to CUDA via C++. We observe this makes PyTorch no less productive for end-users.
>
> We could of course attempt to rewrite the simulator using an array-based programming language like PyTorch or Jax. However, implementing complex training environments using state-of-the-art simulation methods requires complex data structures and non-trivial control flow (traversing acceleration structures, collision solvers, state machines, conditional logic in functions) is cumbersome in array-based abstractions. For this reason, Jax and Torch-based environments rarely contain all of our features while meeting our performance targets.
>
> ### Questions
> - Pedestrian and Cyclist Behavior: **Both pedestrians and cyclists are controlled** within the simulator. We use the same dynamic models as for vehicles, with smaller bounding boxes for pedestrians to reflect realistic behavior.
> - Ethical Statement: $\color{green}{\text{The ethical statement has been added as Section 7 in the current version}}$.
> - Log Scale for Fig. 5: We have the log plot. $\color{green}{\text{Please see the updated Figure 5 in the paper}}$.
>
> ### Conclusion
> Thank you again for your thoughtful comments! We hope we have addressed all of your questions and welcome any further discussion. If all concerns are resolved, we kindly ask that you consider increasing your support for our paper.

---

> > ### Comment · Reviewer_AUkE · 2024-11-27
> >
> > Thanking the authors for addressing the questions.
> > It would be nice if the pending (orange colored) tasks are also completed and get mentioned in the final version of the paper.
> > I went through the comments and discussions from other reviewers as well and it seems that the authors have tried to carefully address their concerns as well.
> > Appreciation for the authors was putting efforts in this direction and releasing their work. This work is a good combination of research and practical engineering. I would like to maintain my score considering the limitations pointed out by some other reviewers.
> > Best wishes! :)

---

### Author Response · Authors · 2024-11-15
**Main response**

We would like to thank all the reviewers for their time and constructive feedback. We address each comment below and highlight the $\color{green}{\text{feedback that is directly incorporated in green}}$ and items we are $\color{orange}{\text{actively working on in orange}}$.

We hope that the reviewers feel we have addressed their questions and welcome further discussion.

---

> ### Author Response · Authors · 2024-11-21
> **Checking in**
>
> Dear reviewers,
>
> As the discussion period is coming to an end, we kindly ask for your engagement with our rebuttal. We have put significant effort into addressing your concerns and would greatly appreciate any further feedback or discussion.
>
> Thank you all for the time and thoughtful comments so far!

---

### Meta-Review · Area_Chair_PuDM · 2024-12-18

**Metareview:**

The authors propose GPUDrive, a novel GPU-accelerated high fidelity simulator designed for multi-agent reinforcement learning in autonomous driving. This high-performance simulation environment allows for efficient training of reinforcement learning agents in complex, multi-agent scenarios. Furthermore, the authors highlight the simulator's ability to enable agents to navigate thousands of scenarios within hours.

### Strengths
- Novelty: GPUDrive introduces a new approach to autonomous driving simulation that leverages the power of GPUs for high-performance and scalability.
- Practical relevance: The simulator provides a realistic and efficient training environment for autonomous driving, whose efficiacy is clear from experiments on real-world data from Waymo OMD.
- Scalability: GPUDrive can handle a large number of agents and complex scenarios, making it suitable for studying intricate interactions in traffic.
- Open-source: The availability of the code base and pre-trained agents promotes further research and development in the field.

### Weaknesses

- The evaluation of GPUDrive seems limited to Waymo Open Motion dataset and IPPO algorithm. Further evaluation on a wider range of scenarios and different types of algorithms could strengthen the paper's claims. Additionally, improving the diversity of simulation agents seem to be critical.

- Empirical comparisons with state-of-the-art simulators other than Nocturne (e.g., Waymax) in terms of simulation speed, realism, and scalability would provide a better context for GPUDrive's performance and capabilities.

 - The computational cost of running high-fidelity simulations in GPUDrive with a large number of agents could be significant. This could limit the accessibility of the simulator for researchers with limited computational resources.

- While the authors provide the code base, providing detailed documentation on how to install, configure, and use the simulator, along with clear explanations of the code and its functionalities, would facilitate wider adoption of GPUDrive.

Decisions to accept / reject: This simulator can accelerate the development and testing of autonomous driving algorithms, as it allows researchers to evaluate the performance of their agents in a wide range of situations efficiently. Despite weaknesses pointed above, this work would be a useful addition to the conference. As such, I'm recommending conditional acceptance assuming the authors would address the above weaknesses.

**Additional Comments On Reviewer Discussion:**

While most reviewers were in favor of acceptance, one reviewer had several concerns, such as lack of domain randomization, presentation issues, and lack of experiments with additional algorithms other than IPPO. The author comments addressed the issue of domain randomization but other issues remain unaddressed. These issues could potentially be addressed in the revision and I highly recommend the authors to do so.

---

### Decision · Program_Chairs · 2025-01-22

Accept (Poster)